# Barley, Triticale, or Rye? The Type of Grain Can Affect the Growth Performance and Meat Quality of Sustainable Raised Pigs

**DOI:** 10.3390/ani13081331

**Published:** 2023-04-13

**Authors:** Eugeniusz R. Grela, Edyta Kowalczuk-Vasilev, Małgorzata Świątkiewicz, Grzegorz Skiba

**Affiliations:** 1Institute of Animal Nutrition and Bromatology, University of Life Sciences in Lublin, Akademicka St. 13, 20-950 Lublin, Poland; eugeniusz.grela@up.lublin.pl (E.R.G.);; 2Department of Animal Nutrition and Feed Science, National Research Institute of Animal Production, Krakowska St. 1, 32-083 Balice, Poland; 3The Kielanowski Institute of Animal Physiology and Nutrition, Polish Academy of Sciences, Instytucka St. 3, 05-110 Jabłonna, Poland

**Keywords:** pigs, sustainable feeding, cereal grains, performance, carcass traits, meat quality, fatty acid composition

## Abstract

**Simple Summary:**

Modern pig production seeks to achieve favorable growth results while paying close attention to animal welfare and the quality of the pork, as well as maximizing the use of feed grown on the farm. Sustainability farming can be a production system that meets these tasks. The objective of this study was to determine the effect of high shares of different grain types (barley, rye, or triticale) in the diet of growing-finishing pigs on growth performance, carcass characteristics, meat quality, and fatty acid profile in meat and fat. The 100-day experiment involved 72 pigs assigned into three groups differing in the main base component of the feed mixture: I—barley, II—triticale, III—rye. The obtained results showed quite differentiated influence of the kind of grain as a base in feeding on the production results and meat quality. The use of triticale or barley may provide better growth efficiency of pigs, better nutrient digestibility, and more beneficial fatty acids profile in respect of the health-promoting value of meat, when compared with feeding pigs with rye. Meat of pigs fed with high rye shared feed mixture characterized of lower cholesterol content, better water holding capacity, and higher saturated fatty acids profile, wherein the later one indicated better resistance to oxidation during storage and longer meat shelf life. The rye may be more beneficial for the production of long-matured or traditional meat products.

**Abstract:**

The effect of high level of barley, triticale, or rye as base of complete mixtures for growing-finishing pigs on growth performance, carcass traits, meat quality, and fatty acid profile in meat and backfat was investigated. The 100-day experiment involved 72 pigs, assigned into three groups (24 animals each). Pigs of each group were kept in six pens (two gilts and two barrows per pen). Diets offered to pigs differed in the proportion of cereals as the leading ingredients in the mixture formulation: I—barley, II—triticale, III—rye. The results showed diversified grain influence on the production results and meat quality. Triticale- and barley-based diets ensured better weight gain and lower carcass fatness than rye (*p* ≤ 0.05). The basic nutrients digestibility of mixtures containing triticale was comparable to that containing barley and higher than that of rye (*p* ≤ 0.05). The meat and backfat of pigs receiving diet with triticale or barley was characterized by more favorable fatty acids profile in respect to the health-promoting indicators (atherogenicity and thrombogenicity indexes; hypocholesterolemic/hypercholesterolemic ratio). The cholesterol level in various tissues was the lowest in pigs fed with rye diet, and their meat characterized by better water holding capacity and more SFA. Higher fat saturation indicate better resistance to oxidation during storage and longer meat shelf life. It seems that the supplementation of triticale to diet may improve the growth efficiency of pigs and the health-promoting value of meat, while the supplementation of rye may be better for the production of traditional or long-matured meat products.

## 1. Introduction

The growing human population requires an increasing supply of food, including meat and other products of animal origin. These expectations are covered by the intensive production of farm animals so far. It is our aim to achieve maximal production effects, determined by the number of fatteners sold, short fattening period and high feed utilization. However, the up-to-date knowledge of environmental protection, the consumers’ approach, and guidelines of the “Green Deal” program currently being introduced in the EU, emphasis is placed on the sustainability goals concerning animal living conditions and environmental impact of animal production, paying increased attention to animal welfare, environmental protection (less carbon footprint), and the quality of pork produced [1,2]. Sustainable or circular farming can be a production system fulfilling these tasks. The aim of this kind of farming is to create integrated environmentally and economically sustainable agricultural production systems, which enhance biodiversity, biological cycles, soil biological activity, and shortening of transport [3]. The animal nutrition should be focused on using mostly farm-derived or local feeds and the diet should ensure quality production of the animals rather than maximizing production [4]. Nonetheless, due to the growing awareness of scientists and consumers, there is an increasing interest in this type of food production.

The proper way to increase the crops’ biodiversity and the use of local feeds is to abandon the feeding of pigs with mixtures based on corn and soybean meal, and to introduce a varied diet composed of various types of cereals and native legume seeds as a source of protein. The basic component of mixtures for pigs are cereals which make up the majority of the dose (60–85% share): barley, corn, wheat, triticale, oat, and rye. Keeping in mind that the cost of feeding is about 70% of the total cost of pork production, it is important to pay attention to what cereal is used in balancing feed for pigs. Examination of the possibility of using various cereals in pig feeding is also important from the point of view of the need to shorten supply transport and use the grain currently available locally. Worldwide corn is the most popular grain in pig feeding and the other cereals differ significantly. Triticale, barley, and rye grains contain more crude protein and a better amino acid profile than corn, including higher content of lysine, methionine, threonine, and tryptophan [5]. On the other hand, these grains contain more crude fiber [5] and there are some issues with the presence of anti-nutritional factors such as non-starch polysaccharide (NSP)—mostly β-glucans, pentosans (xylans, arabinans), as well as phytates, and alkyloresorcinols [6,7,8,9]. Barley contains β-glucans in amounts higher than other grains, which lowers its digestibility; however, the carcass of pigs fed with barley is characterized by whiter and firmer fat which is beneficial for high quality pork products. Triticale is a cereal grain species derived from the hybridization of wheat and rye, developed to combine favorable traits from both these grains—growth vigor, cold weather tolerance, and high protein content [10]. Rye is a cereal that can provide a valuable source of energy and protein when included in swine diets. Its nutrient value for protein and energy is similar to wheat and barley; nevertheless, livestock producers used to be reluctant to rye as a feedstuff because of the presence of alkaloids and the anti-nutritional effect of pentosanes which could lead to potentially lower feed intake. However, the modern rye varieties appeared as cost-effective feed for growing pigs, yielding significantly better outcomes on carcass quality in comparison to barley-containing diets [11]. The latter aspects are extremely important in deteriorating climatic and environmental conditions, because they allow limiting fertilization and are a chance to obtain yields in poor soil-environmental conditions. Due to low thermal requirements of rye its losses during winter are small (1–2%), which is especially important in north-eastern regions, because winter snow cover occurs less and less often [12]. Furthermore, the results of research carried out in recent years indicate that the modern rye varieties show some beneficial effects on the intestinal health, modulation of the immune response, and reduction in diarrhea and Salmonella fecal counts [13,14]. Rye contains more crude protein and amino acids than corn, but the standardized ileal digestibility of rye’s amino acids is lower due to higher NSP presence, and finally, the concentration of standardized ileal digestible amino acids is similar in both grains [15]. As an element of sustainable pig production, rye is also worth increased interest because it needs less fertilizer, demands substantially less water, reduces CO_2_ emissions in pork production in comparison with wheat, and can reduce the feed cost in comparison to other cereals [14].

Sustainable feeding is a challenge in pig production, because it affects growth rate, carcass traits, and meat quality in several different ways, depending on environmental conditions, local feed resources, and the composition of feed rations [16,17]. As the sustainable and circular production system differs markedly from the conventional pig production and the effects on overall meat quality need to be studied. Some research results presented the deteriorative effect of organic feeding on the quality traits of carcass (meat or ham percentage, backfat thickness), as well as meat (drip loss, shear force, tenderness) [3], whereas some studies showed improved meat juiciness or dietetic value (higher PUFA n3 content) in pigs fed according to the organic system [18]. Therefore, the hypothesis tested in the present experiment was that in the growing-finishing pigs, the cereal type itself may differentiate production effects and pork quality. The objective of our study was to determine the effect of the high share of barley, rye, or triticale in growing-finishing pigs on growth performance, carcass traits, meat quality, and fatty acid profile of meat fat and backfat.

## 2. Materials and Methods

### 2.1. Ethical Approval

The experiment was conducted according to the guidelines of the Declaration of Helsinki and in compliance with the European Union law (*Directive 2010/63/UE, received in Poland by Legislative Decree 266/2015*) of the European Parliament and of the Council on the Protection of Animals Used for Scientific or Educational purposes. Ethical approval was issued by the Local Ethics Committee in Lublin, Poland—Resolution No 33/2010 of 16 February 2010.

### 2.2. Experimental Design, Animals, Housing, and Feeding

The experiment involved 72 crossbred pigs (Polish Large White × Polish Landrace) with an initial body weight of 30.0 ± 0.5 kg. Pigs of each group were kept in 6 pens (2 gilts and 2 barrows per pen). Diets offered to pigs differed in the proportion of cereals as the leading ingredients in the mixture formulation: group I—barley, group II—triticale, and III—rye. All cereals used in the study were produced organically by producers running organic farms. During the study, animals of each group were fed diets with energy, total protein, amino acid, minerals, and vitamins content balanced to their age [5]: starter diet (30–55 kg BW—35 days), grower diet (56–87 kg BW—35 days), and finisher diet (78–115 kg BW—30 days) (Table 1). The pigs were fattened for 100 days. Feed intake was controlled per pen, by weighing the amount of feed poured into the feeder and the amount of leftovers. Although the animals were kept in group pens, individual body weight control was performed and the daily weight gains also were calculated individually. Animals were weighed 4 times (at the 1st, 35th, 70th, and 100th day of experiment).

All animals had free access to feed and water with access to an outdoor yard of 5 square meters per head. The environmental conditions (temperature, relative humidity, and cooling) were the same for all the groups.

### 2.3. Tissue Collection

After 100 experimental days of the body weight averaging 113 ± 3.0 kg, all pigs were transported to an abattoir, following a 12 h fast. The slaughter procedure was conducted in accordance with the technology currently employed in the meat industry, using the electrical stunning. Carcass were weighed and the content of meat in the carcass was calculated. The content of lean meat in the carcass was determined using the Ultra FOM 300 apparatus (SFK Technology A/S, Herlev, Denmark) within 30 min from the start of slaughter operations. Fat thickness was estimated over the shoulder, on the back, and in three sites on the os sacrum area. Samples of muscle *longissimus lumborum* (MLL) and backfat were collected 24 h after slaughter from 12 animals of each group. The tissues were sampled from the carcasses of 2 pigs of an average body weight within each pen (112.1 ± 0.5 kg in group I, 114.3 ± 0.6 kg in group II, and 111.2 ± 0.4 kg in group III). The MLL samples were taken from near the last thoracic and first lumbar vertebra area. Backfat was sampled from above the scapula by cutting out a slab about 6 cm wide and 20 cm long. Immediately after collection, the meat and backfat samples were stored individually in plastic bags at about −20 °C.

### 2.4. Ileal Digestibility

For the ileal digestibility coefficient of nutrients, the indicator method analysis with silica (SiO_2_) as a marker was used. Immediately after slaughter (pigs were not fasted prior to slaughter and collection of the sampling of intestinal digesta), approximately 100 mL of digesta were collected from the last 100 cm of the distal ileum and stored at −20 °C for subsequent determination of ileal digestibility. Samples of ileal digesta were lyophilized, and along with samples of diets, finely ground to pass through a 1 mm screen prior to the chemical analyses. Ileal digesta samples were analyzed, using the AOAC procedures [20], for DM, CP, EE, CF, and NSP as explained for diet samples. The silica was determined according to [20]’s method.

Apparent ileal digestibility coefficients (AIDC) were calculated using the following equation:AIDC (%) = 100 − (100 × (a × b)/(c × d)
where:

a—the silica content in the feed (%)

b—the nutrient content in digesta (%)

c—the silica content in digesta (%)

d—the nutrient content in feed (%)

### 2.5. Meat and Fat Quality Analysis

Physicochemical properties pH_1_ and pH_2_, electrical conductivity, meat color, and water holding capacity were determined in the samples of lumborum muscle from the lumbar section (MLL), according to the methods described by [21] Grela et al. (2020).

Lipid quality indices, i.e., atherogenicity index (AI) and thrombogenicity index (TI), were calculated according to the [22] Ulbricht and Southgate (1991) equations:AI = [(4 × C14:0) + C16:0]/[n6 PUFA + n3 PUFA + MUFA]
TI = [C14:0 + C16:0 + C18:0]/[(0.5 × MUFA) + (0.5 × n6 PUFA) + (3 × n3 PUFA) + n3/n6 PUFA]

Hypocholesterolemic/Hypercholesterolemic ratio (h/H) was calculated according to [23] Fernández et al. (2007)’s formula:h/H = (C18:1 + C18:2 + C18:3 + C20:3 + C20:4 + C20:5 + C22:4 + C22:5 + C22:6)/(C14:0 + C16:0)

### 2.6. Chemical Analysis

The cereal grains and diet samples were determined for contents of basic nutrients according to standard AOAC procedures [20]. Calcium content was determined in an ASA SOLAR 939 UNICAM flame spectrophotometer, whereas phosphorus content was measured with the spectrometric method according to [20]. Total lysine was determined with ion-exchange chromatography in a 119 Cl Beckman amino acid analyzer (Beckman Instrument Company, Brea, CA, USA). Before analyses, the samples were subjected to acidic hydrolysis in the presence of 6 M HCl, at a temp. of 105 °C for 24 h. Sulfuric amino acids were determined separately after oxidation [24]. Standard methods were used for quantification of anti-nutritional compounds such as phytic acid [25] and total polyphenols [26]. The content of NSP total in cereals and mixtures was determined by the enzymatic method according to Englyst et al. (1982) [27]. Content of total resorcinol with side chains from C13:0 to C27 was isolated from mature barley, wheat, and rye grains according to the method described by Kozubek (1985) [28].

Total lipids for the fatty acid analysis were extracted from the backfat and MLL with a chloroform/methanol mixture, according to Folch et al. (1957) [29]. A percentage of fatty acid methyl esters was estimated using the gas chromatography procedure on a Varian CP-3800 chromatograph described by Grela et al. (2020) [21]. Cholesterol content in organs and tissues was measured using the colorimetric method of Rhee et al. (1982) [30].

### 2.7. Statistical Analysis

The obtained data were analyzed statistically using a general linear model (GLM) of one-way analysis of variance (ANOVA) using Statistica 12 software ver. 10 [31]. Duncan’s test was applied for multiple comparisons among the means. Differences were considered as significant at *p* < 0.05, whereas *p* value between 0.05 and 0.10 was considered a trend/tendency. The pen served as the experimental unit for feed intake and feed conversion ratio (*n* = 6 per group). For all indices which were measured for each animal separately, the individual pig served as the experimental unit (body weight, average daily gain, digestibility of nutrients, carcass and meat traits, and fatty acids composition in tissues) (*n* = 24 per group). All meat quality indices and fatty acid composition in tissues were analyzed for 8 pigs per group (4 gilts and 4 barrows each).

## 3. Results

The analysis of the chemical composition of the particular organically produced cereals used in the study as the leading cereal in the mixture formulation revealed varying content of total protein and other nutrient contents, as well as varying fatty acid profiles (Table 2). The highest amount of protein was found in triticale grain and the least in rye. The highest crude fiber, NDF, ADF, and total NSP were recorded in barley grain. The rye grain was characterized by the lowest content of crude protein, crude fat, including unsaturated fatty acids. The highest content of lysine and sulphuric amino acids was noted in triticale grain. The highest content of tannins and alkylresorcinols was found in rye grain, while phytates were found in barley.

The production results are summarized in Table 3. The lowest slaughter weight, and thus, the lowest daily gains during the whole 100-day fattening period, were obtained in group III, for which rye was the main grain component of the mixture. The best daily weight gains were observed in group II (851 vs. 823 and 806 g), which received triticale in the mixture. However, in both cases, a statistically significant difference was noticed between group III and II only (*p* ≤ 0.05). Daily feed intake was similar among groups in starter and grower periods, with a decreasing tendency (*p* = 0.063) in group III (rye) in finisher period. Feed conversion ratio throughout the fattening period was similar in all groups.

Apparent ileal digestibility coefficients (AIDC) of crude protein, ether extract, nitrogen-free extracts, and NSP did not differ significantly between group II (triticale) and I (barley) and were significantly higher than in group III (rye) (Table 4). A similar tendency was noticed in AIDC of dry matter and crude fiber (*p* = 0.098 and 0.066, respectively).

There were no significant differences in slaughter performance and meatiness indices (Table 5), but a tendency toward higher meat content in ham (*p* = 0.063) was observed in fatteners of group II (triticale). A significantly thicker backfat (*p* ≤ 0.05) was recorded in the fatteners of group III (rye), while the differences between group II (triticale) and I (barley) were statistically insignificant.

The cholesterol content in different tissues of pigs is presented in Table 6. The lowest content of cholesterol in meat, fat, and liver were found in the pigs of group III (rye), compared to remaining groups (*p* ≤ 0.05). However, the cholesterol content in meat and fat in groups I (barley) and II (triticale) was similar. The cholesterol content differed significantly only between groups I (barley) and III (rye) (*p* ≤ 0.05).

The indicators of meat quality (*longissimus* m.), such as pH and electrical conductivity, did not differ significantly among the experimental groups (Table 7). Larger differences were observed in meat color, as the lightest color (*p* = 0.037) characterized the meat of group III (rye), while the highest yellowness (*p* = 0.044) was observed in the meat of group II (triticale). Meat from pigs of group III (rye) showed the lowest values of the water holding capacity index (G-H, mg) (*p* = 0.041) and the lowest infiltration area (G-H, cm^2^) (*p* = 0.045). The highest value of M/T parameter was noticed in meat of pigs fed mixture containing rye as a main cereal ingredient (*p* = 0.039).

The fatty acid content in MLL meat is shown in Table 8. The lowest (*p* = 0.036) SFA content was found in group II (triticale), but the statistical difference was confirmed only between group II (triticale) and group III (rye). The highest MUFA (*p* = 0.029) and PUFA (*p* = 0.042) contents were found in the meat of group II (triticale). The meat of these pigs was also characterized by the lowest PUFA n6/n3 ratio (*p* = 0.044) and, also, the most favorable values of AI (*p* = 0.041), TI (*p* = 0.045), and h/H (*p* = 0.039) indices.

The fatty acids composition in the adipose tissue is presented in Table 9. In the backfat, similar relationships in the fatty acid profile were observed as in the meat, but significant differences were found only in the SFA (*p* = 0.031) and MUFA (*p* = 0.038). A tendency to the most favorable, from a dietary point of view, fat quality indices were found in the backfat of pigs of the group II (triticale): PUFA n6/n3 (*p* = 0.074), AI (*p* = 0.057), TI (*p* = 0.053), h/H (*p* = 0.061).

## 4. Discussion

Due to the negative effects on the environment and animal welfare caused by the intensive conventional pig production, the increasing consumers’ interest in pork from sustainability and low-input production systems was observed in recent years. However, the variability of the composition of cereals from ecological/organic crops, especially limited availability of the essential amino acids [32], can considerably restrict the possibilities for the optimal adaptation of the diets to the specific requirements of growing pigs. Thus, there is a concern that nutritional imbalances encountered in practice might lead to deterioration of the growth rate [12] as well as in pork quality of such fed pigs. In the present study, fatteners were fed with the most popular barley, as a control group, while the other two groups included triticale and rye. The literature data concerning influence of different type of cereals from organic production on growth performance, carcass traits, meat and fat quality of fattening pigs are sparse due to the low availability of cereals from this type of cultivation. Thus, a great part of this discussion is based on the results of studies where varied types of cereals were used; however, they were obtained from conventional cultivation.

The literature’s data showed that dietary energy and protein/amino acids content are key drivers of the voluntary feed intake in pigs and, therefore, daily gain, depending on the type of cereal dominating in the dose [33,34,35,36,37]. Beaulieu et al. (2009) [38] showed that a response of pigs to lowering of the dietary available energy increased feed intake in order to maintain a constant daily energy intake. However, this ability to compensate for lower energy density is not always observed, especially when the maximum feed intake is limited by the ingestion capacity of the pigs [39]. In the presented study, the observation was that the kind of the used grain had no significant effect on feed intake and feed conversion ratio. The reason could be similar energy and nutrient content in all used diets. In the present study, despite no difference in feed consumption and feed conversion, the grower and finisher pigs fed diet with triticale as a main cereals component grew faster than pigs fed with barley or rye. This could be due to lower presence of the non-starch polysaccharides and other anti-nutritional factors in triticale compared to rye and barley [40], which was also confirmed by our chemical analysis. However, Chuppava et al. (2020) [14], in the study with young pigs fed diets with higher proportions of rye, observed that feed intake, average daily gain, and feed conversion ratio were similar to animals fed diets with high amounts of wheat. Additionally, [41] Grone (2018) and [42] Wilke (2020) observed no negative impact on growth performance when young pigs were fed diets containing a high proportion of rye. In turn, in the study by [43] Turyk et al. (2011), the pigs fed triticale based diets showed higher growth rate and better feed conversion ratio, which was contrary to [44] Sullivan et al. (2007), who observed deterioration of weight gains and feed conversion in pigs obtaining 80% of triticale in feed mixture, although the feed intake was similar among groups.

In the present study, higher digestibility of protein, and also fat, in pigs fed the triticale- or barley-based diet was observed, while the lowest coefficients were noticed in pigs fed diet based on rye. However, the literature data concerning digestibility of nutrients in mixture based on triticale and rye are ambiguous. McGhee and Stein (2020) [15], in the study with growing pigs, found lower digestibility of energy in rye-fed pigs compared to pigs fed with wheat, corn, and sorghum. They highlighted that feed based on rye was more intensively fermented (its non-starch polysaccharides) in the hindgut, due to the dietary fiber consists of arabinoxylan, fructooligosaccharides, cellulose, and mixed-linked ß-glucans [13]. However, authors did not use feed based on triticale, they found that pigs fed diet based on rye consumed less feed/energy than pigs fed with barley, but similar to pigs fed with wheat or corn diet. Burbach et al. (2017) [45] showed a lower digestibility for rye bread, which is in agreement with reduced digestibility of crude protein from rye compared to triticale in growing pigs [46,47,48]. One of the reasons of lower nutrients digestibility in animals fed with mixture including high amount of rye could be an increased viscosity of the ileal digesta [49], which impairs the capacity of digestive enzymes to degrade substrates. In turn, a reduction in available nutrients might also initiate changes in the composition of the intestinal microbiota. Contrarily, the results of the study by Thacker et al. (1999) [50] showed that the rye-based diets had higher digestibility coefficients for dry matter, crude protein, and gross energy than barley, when rye was included in a young growing pig diet at 60%.

The results of the presented experiment indicated that the cereals had no effect on slaughter yield and indicators of the meatiness, but use of diet based on rye increased backfat thickness of pigs. Some research data [51] showed that replacement of barley with rye into diets for growing-finishing pigs resulted in increasing of carcass weight and slaughter value. In turn, in the study by Bussières (2018) [51], in which barley was replaced with rye, the only effect was reduction in backfat thickness. Sullivan et al. (2007) [44] noticed deteriorative effects of 80% of triticale in feed mixture (replacing corn) on loin meat area, but no effect on the backfat thickness of carcass. Generally, it seems that inclusion of rye into diets for growing-finishing pigs does not influence growth performance and carcass indicators, but the effect of growth stage (age of pigs) may appear. According to McGhee et al. (2021) [52], pigs fed a diet based on rye might consume less feed in the grower phase, which results in lower weight gains, but growth performance for the entire fattening period and carcass quality remain unaffected. This might be due to volatile compounds associated with bitter flavors present in rye grain. In addition, the soluble dietary fiber fractions in rye increase the viscosity of digesta which may increase gut fill and give the satiety feeling in monogastric animals—as a consequence, it may be a reason for the lower feed intake tendency (Kristensen and Jensen, 2011) [53]. More desirable slaughter parameters in pigs fed triticale, compared to animals fed barley, were presented by Turyk et al. (2011) [43]. Authors found that carcass of pigs fed diet based on triticale was characterized by higher loin eye area and smaller carcass’s fat weight, while parameters such as dressing percentage, carcass length, fleshiness, and backfat thickness were not affected by type of cereals added to feed. Thus, it generally seems, in the present experiment, that the grain type in diets pigs does not influence slaughter yield.

Color, as an indicator of meat freshness when it is light and red, is the most popular attribute of consumers when making the purchasing decision [52]. The results of the present study indicated that type of cereals in diet influenced meat color indices, as rye diet increases meat lightness, while triticale diet increases yellowness. The lighter color of loin meat and backfat observed for pigs fed with rye is in agreement with effects observed by Kim et al. (2014) [54], when including barley instead of corn in diets for finishing pigs. It is known that corn contains more carotenoids than other cereals grains, which seems to be an explanation for that increase in color lightness. However, Sullivan et al. (2007) [44] did not notice any effect on meat color parameters (L*, a*, b*) when triticale replaced the corn in pig diet. In our case, it is difficult to explain the differences in the meat color with the carotenoid content in the grain, because corn was not used, but another factor influencing the meat and fat color may be the fatty acids profile. Fat cells containing saturated fat with a high melting point appear whiter than when unsaturated fat with a lower melting point is present (Burnett et al., 2020) [55]. This is consistent with the observation in our present experiment that meat from pigs fed with rye characterized by the highest SFA content and the highest lightness. In addition, PUFA (ALA, EPA, and DHA) are susceptible to oxidation; so, the higher the content of these acids in tissue, the higher content of oxidation products. The oxidation products lead to darker meat by a reduction in redness, increased hue angle or color saturation [55]. In the present experiment, the meat from pigs fed with triticale contained the highest amount of PUFA (*p* ≤ 0.05), which may explain why their meat was the darkest. Its hue angle and color saturation were higher when compared to meat from pigs fed with barley or rye; however, these differences were not statistically significant. It is compatible with the experiment by Turyk et al. (2011) [43] in which the researchers noticed that semimembranosus muscle of pigs fed with triticale had a slightly darker color than pigs fed with barley. The effect of fatty acids on meat color in pigs is not so significant as in beef but must be taken into consideration in organic farming which is dedicated to the production of high-quality meat products.

In the present experiment, diet based on rye significantly improved the water holding capacity and the parameter measured (G-H cm^2^) was lower by about 8% in comparison to group obtaining barley or triticale. McGhee et al. (2021) [52] did not observe any effect of rye on water properties of pig meat measured as drip loss and cooking loss. Likewise, drip loss was not impacted by inclusion of rye in the pig diets (25% in grower and 50% in finisher phase) [51], as well as triticale (80%) [44]. However, diet for broiler chickens based on rye is able to significantly ameliorate the water holding capacity of meat [56]. In the experiment by [44] Turyk et al. (2011), meat of pigs fed with triticale was characterized by higher water holding capacity when compared to meat obtained from pigs fed with barley. However, in the present experiment, there was no significant difference between water holding capacity of meat from pigs fed with barley or triticale (7.96 vs. 7.93 cm^2^).

The nutritive quality of investigated tissues was determined also by the content of SFA, MUFA, and PUFA, as well as their health quality, by calculating the atherogenic (AI) and thrombogenic indices (TI). The AI index shows a relationship between SFA (pro-atherogenic), which promote the attachment of lipids to cardiovascular endothelial cells, and PUFA (anti-atherogenic), which reduce cholesterol levels and prevent the occurrence of coronary artery disease. The TI index indicates a tendency for clots forming in blood vessels. The results of the present study indicate that feed based on triticale decreased, while diet based on rye increased, SFA content in the meat and backfat. The opposite effect was found for MUFA and PUFA. Moreover, it also seemed that triticale-based diet improves, while rye diet deteriorates, the ratio of PUFA n6/n3, AI, TI, and h/H indexes—both in meat (*p* ≤ 0.05) and in backfat (*p* > 0.05). In the study of Sullivan et al. (2007) [44], the meat fatty acids profile of pigs fed with triticale (80% of diet) instead of corn was not affected. In the case of triticale fed pigs (as replacement for barley), there were observed somewhat better dietary properties of meat due to more beneficial fatty acid profile—more PUFA n3 and lower PUFA n6/n3 ratio [57]. The ratio of PUFA n6/n3 is a risk factor in cancers and cardiovascular disease and a lower PUFA n6/n3 ratio is required for the prevention and management of chronic diseases. Currently, consumers are aware of the health benefits and nutritional quality of meat. There are research reports that consumers are willing to pay more for healthy food, such as organic food, including PUFA n3-enhanced meat [55]. Consumers’ expectations cause that the interest in the possibilities of manipulating the fatty acid composition towards more favorable profile is growing. The results of our experiment support the idea that choosing the proper grain for pig feed mixture can be a way of obtaining meat products with health-promoting properties.

Fatty acids composition can also affect palatability and various technological traits of meat quality. MUFA and the main in this group, oleic acid, are positively correlated with consumer flavor preference, juiciness, tenderness, and overall meat liking (Burnett et al., 2020) [56]. In our study, triticale diet may have a potential to produce pork with significantly higher palatability than rye, but which is comparable with barley. On the other hand, the impact of fatty acids on the meat-eating score is not clear and taste may change significantly during storage or technological processes. The fatty acids have a serious effect on firmness or softness of the fat in meat because of differ melting points. The ability of unsaturated fatty acids, especially PUFA n3, to rapidly oxidize may negatively indicates the shelf life of meat and loin-fat tissues separation in bacon or ham. In case of high quality long-maturing products, these technological traits are extremely important. It is known that oxidation of PUFA yielded numerous volatile compounds—aldehydes, alcohols, furans, and ketones [58]—detectable for consumers. The results of the present experiment indicated that feeding pigs with rye, as a main grain, allows to obtain meat that is more resistant to oxidation, and thus, better retaining freshness and palatability values during storage. Further research is needed to test the effect of various grains in pigs diet on the meat palatability changes during storage.

## 5. Conclusions

The results of the experiment showed quite a diversified influence of the grain type in the diet on the production results and meat quality of pigs, and no influence on the carcass cold dressing yield and meatiness. Triticale- and barley-based diets ensured better body weight gain and feed conversion, as well as lower carcass fatness, when compared to rye. The digestibility of nutrients (especially crude protein and fat) in feed mixtures containing triticale was comparable to that of barley, and both were higher than in diets based on rye. The meat of pigs receiving diet based on triticale was characterized by more favorable values of health-promoting parameters: PUFA n6/n3 ratio, AI, TI, and h/H. On the other hand, meat from pigs fed with rye was characterized by lighter color, better water holding capacity, and a higher content of SFA. The last two indicators are very important for good palatability, as well as oxidative stability and the shelf life of meat. In addition, the cholesterol level tested in various tissues (meat, fat, liver) was the lowest in pigs fed with feed based on rye. In summary, our experiment showed how different grains available locally can affect fattening results and meat quality differently. On the other hand, our results showed that choice of grain for pig feed mixtures may depend on the local purpose of production: the use of triticale instead of barley may improve the fattening efficiency of pigs and the health-promoting value of meat, while the use of diet based on rye may be better for the production of traditional or dry-cured/long maturing meat products.

## Figures and Tables

**Table 1 animals-13-01331-t001:** Composition (%) and nutritive value of fattening pig diets.

Feeding Period	Starter (35 Days)			Grower (35 Days)			Finisher (30 Days)		
Groups	I Barley	II Triticale	III Rye	I Barley	II Triticale	III Rye	I Barley	II Triticale	III Rye
Wheat (Astoria)	19.16	19.86	19.16	10.0	10.0	10.0	0	0	0
Barley (Basic)	60.0	0	0	67.0	0	0	80.0	0	0
Triticale (Tulus)	0	60.0	0	0	67.0	0	0	80.0	0
Rye (Tur)	0	0	60.0	0	0	67.0	0	0	80.0
Soybean extruded (Mavka)	4.2	4.0	4.2	3.0	3.0	3.0	2.0	2.0	2.0
Wheat bran	0	0	0	10.11	10.41	10.11	14.35	14.55	14.35
Yellow lupine (Perkoz)	4.0	3.8	4.0	4.0	3.8	4.0	0	0	0
Calcium carbonate	1.2	1.2	1.2	1.2	1.2	1.2	1.2	1.2	1.2
Dicalcium phosphate	0.8	0.7	0.8	0.1	0.1	0.1	0	0	0
Sodium chloride	0.4	0.4	0.4	0.4	0.4	0.4	0.4	0.4	0.4
Complementary mix ^1^	1.2	1.2	1.2	1.05	1.05	1.05	0.9	0.9	0.9
Yeast	8.8	8.6	8.8	3.0	2.9	3.0	1.0	0.8	1.0
Lysine	0.2	0.2	0.2	0.12	0.12	0.12	0.15	0.15	0.15
DL-methionine	0.04	0.04	0.04	0.02	0.02	0.02	0	0	0
Content in 1 kg:									
ME, MJ/kg	13.08	13.37	13.14	12.79	12.84	12.72	12.49	12.54	12.48
Dry matter, g	892.4	891.6	892.1	889.7	891.2	888.9	889.6	890.4	891.2
Crude ash, g	44.5	42.8	43.1	43.6	41.9	41.4	41.6	40.8	40.7
Crude protein, g	171.1	171.8	170.2	151.6	152.4	150.5	130.1	133.2	130.8
Crude fiber, g	41.5	39.8	38.9	44.5	43.8	43.3	46.7	46.3	45.7
NDF, g	222.4	219.4	218.2	185.7	183.5	179.5	200.4	194.2	187.8
ADF, g	63.3	59.6	58.8	64.8	62.2	61.8	73.3	72.3	71.8
Ether extract, g	38.5	38.4	38.9	36.4	35.8	35.9	32.7	34.1	31.5
Total Lys, g	8.02	7.98	8.01	7.01	6.92	7.02	6.06	6.08	6.05
Total Met, g	2.74	2.76	2.71	2.33	2.28	2.26	1.91	1.92	1.89
Ca, g	6.60	6.61	6.69	5.53	5.48	5.49	4.82	4.81	4.83
P, g	5.38	5.37	5.41	4.79	4.73	4.69	4.41	4.38	4.32

^1^ Complementary mix Dolmix S RE added per kg of diet: 100 mg Fe (Fe(SO_4_) × 7H_2_O); 100 mg Zn 0(ZnO); 23 mg Cu (CuSO_4_ × 5H_2_O); 1.2 mg I (CaI_2_); 0.3 mg Se (Na_2_SeO_3_); ME—metabolizable energy estimated according to the equation Bayer et al. (2003) [19]; NDF = neutral detergent fiber assayed without heat-stable amylase and expressed inclusive of residual ash; ADF = acid detergent fiber expressed inclusive of residual ash.

**Table 2 animals-13-01331-t002:** Chemical composition of cereals grain, including the anti-nutritional factors content (g/kg).

Item	Barley	Triticale	Rye
Basic Chemical Composition			
Dry matter	884.2	883.8	881.6
Crude ash	27.4	19.8	18.7
Crude protein	116.2	129.5	112.7
Ether extract	20.8	21.3	19.5
Crude fiber	44.7	28.1	25.6
NDF ^1^	182.7	129.4	138.2
ADF ^2^	55.7	32.1	37.6
Lysine	3.78	3.95	3.54
Methionine + cysteine	4.13	4.34	3.87
Palmitic acid (16:0)	2.93	1.12	2.58
Stearic acid (18:0)	0.21	0.11	0.28
Oleic acid (18:1)	2.65	1.73	1.77
Linoleic acid (C18:2)	7.59	8.27	4.92
Linolenic acid (C18:3)	0.76	0.65	0.62
Anti-nutritional factors			
NSP ^3^	162.6	124.7	148.2
Alkylresorcinols	0.102	0.523	1.121
Tannins	1.58	1.41	1.87
Phytate	109.4	89.5	96.3

^1^ NDF—neutral detergent fiber assayed without heat-stable amylase and expressed inclusive of residual ash; ^2^ ADF—acid detergent fiber expressed inclusive of residual ash; ^3^ NSP—non starch polysaccharides.

**Table 3 animals-13-01331-t003:** Productive performance of pigs during the 100 days of fattening experiment.

Item	Groups/Cereal			SEM	*p* Value
	I—Barley	II—Triticale	III—Rye		
Initial body weight, kg	30.2	30.1	30.2	0.11	0.321
Body weight at 35 d, kg	55.8	55.3	54.4	1.25	0.102
Body weight at 70 d, kg	87.6 ^ab^	89.2 ^a^	86.5 ^b^	2.26	0.041
Body weight at 100 d, kg	112.5 ^ab^	115.2 ^a^	110.8 ^b^	1.64	0.042
Average daily weight gains (ADG), g					
Starter period	731 ^a^	720 ^a^	691 ^b^	34.5	0.043
Grower period	909 ^b^	969 ^a^	917 ^b^	78.3	0.014
Finisher period	830 ^b^	867 ^a^	810 ^b^	59.6	0.038
Whole period	823 ^ab^	851 ^a^	806 ^b^	58.7	0.034
Average daily feed intake (DFI), kg					
Starter period	2.02	2.05	1.94	0.02	0.102
Grower period	2.62	2.66	2.56	0.07	0.109
Finisher period	2.98	2.95	2.82	0.09	0.063
Whole period	2.52	2.53	2.42	0.08	0.059
Feed conversion ratio (FCR), kg/kg					
Starter period	2.76	2.85	2.81	0.05	0.097
Grower period	2.88	2.75	2.79	0.05	0.105
Finisher period	3.59	3.40	3.48	0.07	0.123
Whole period	3.07	2.97	3.00	0.06	0.092

^a, b^—values in the same rows with different letters differ significantly at *p* ≤ 0.05.

**Table 4 animals-13-01331-t004:** Apparent ileal digestibility coefficients (%) of nutrients in fattened pigs.

Ingredients	Groups/Cereal			SEM	*p* Value
	I—Barley	II—Triticale	III—Rye		
Dry matter (DM)	76.5	76.8	74.8	0.25	0.098
Crude protein (CP)	77.5 ^a^	78.1 ^a^	73.8 ^b^	0.43	0.035
Ether extract (EE)	81.1 ^ab^	82.3 ^a^	79.9 ^b^	1.21	0.041
Nitrogen-free extracts (NfE)	91.5 ^a^	92.1 ^a^	89.3 ^b^	0.98	0.047
Crude fiber (CF)	31.3	32.7	29.3	0.81	0.066
Non-starch polysaccharides (NSP)	21.4 ^a^	21.9 ^a^	18.2 ^b^	0.52	0.035

^a, b^—values in the same rows with different letters differ significantly at *p* ≤ 0.05.

**Table 5 animals-13-01331-t005:** Carcass quality traits of fattened pigs.

Item	Groups/Cereal			SEM	*p* Value
	I—Barley	II—Triticale	III—Rye		
Cold dressing yield, %	78.8	78.9	79.5	0.25	0.098
Meat of ham, %	78.4	80.2	79.5	0.39	0.063
Loin eye area, cm^2^	52.4	52.9	52.5	0,55	0.115
Meatiness of carcass, %	55.3	55.9	55.2	0.28	0.147
Average backfat thickness from 5 measurements, cm	1.97 ^ab^	1.94 ^b^	2.05 ^a^	0.12	0.046

^a, b^—values in the same rows with different letters differ significantly at *p* ≤ 0.05.

**Table 6 animals-13-01331-t006:** Cholesterol content in different tissues (mg/g) of fattened pigs.

Item	Groups/Cereal			SEM	*p* Value
	I—Barley	II—Triticale	III—Rye		
Meat (*Longissimus lumborum* m.)	0.66 ^a^	0.64 ^a^	0.52 ^b^	0.04	0.044
Adipose tissue (backfat)	1.19 ^a^	1.15 ^a^	1.03 ^b^	0.07	0.048
Organ tissue (liver)	3.31 ^a^	3.18 ^ab^	3.07 ^b^	0.14	0.032

^a, b^—values in the same rows with different letters differ significantly at *p* ≤ 0.05.

**Table 7 animals-13-01331-t007:** Meat quality indices (*longissimus lumborum* m.).

Item	Groups/Cereal			SEM	*p* Value
	I—Barley	II—Triticale	III—Rye		
pH_1_ 45 min after slaughter	6.27	6.25	6.28	0.08	0.196
pH_2_ 24 h after slaughter	5.54	5.55	5.56	0.07	0.217
Electrical conductivity mS/cm	18.3	18.8	18.5	0.38	0.131
Meat color CIE:					
lightness (L)	53.78 ^ab^	52.14 ^b^	55.51 ^a^	1.25	0.037
redness (a)	19.24	19.83	19.07	0.36	0.101
yellowness (b)	1.43 ^b^	1.64 ^a^	1.42 ^b^	0.18	0.044
Chroma (C)	19.54	19.77	19.52	0.41	0.148
Hue angle (H°)	4.7	4.8	4.6	0.66	0.131
Water holding capacity:					
G-H, cm^2^	7.96 ^a^	7.93 ^a^	7.37 ^b^	0.27	0.045
G-H, mg	76.73 ^a^	76.69 ^b^	70.16 ^b^	3.14	0.041
M/T ×100	23.11 ^ab^	21.35 ^b^	25.14 ^a^	1.84	0.039

^a, b^—values in the same rows with different letters differ significantly at *p* ≤ 0.05; M/T—meat sample/total loss × 100; G-H—free water by the Grau–Hamm method.

**Table 8 animals-13-01331-t008:** Fatty acid composition (% of all estimated fatty acid) and indices of fat dietetic value in meat (musculus *longissimus lumborum*).

Fatty Acid	Groups/Cereal			SEM	*p* Value
	I—Barley	II—Triticale	III—Rye		
SFA	41.46 ^ab^	39.93 ^b^	43.43 ^a^	1.42	0.036
C 16:0	24.97 ^ab^	24.16 ^b^	26.45 ^a^	0.11	0.042
C 18:0	14.74	14.05	15.21	0.45	0.054
MUFA	52.13 ^ab^	53.27 ^a^	50.32 ^b^	1.53	0.029
C 18:1 n9	44.02 ^a^	45.02 ^a^	42.34 ^b^	1.32	0.032
C 18:1 n7	3.89	3.95	3.83	0.15	0.238
PUFA	6.22 ^b^	6.61 ^a^	6.11 ^b^	0.17	0.042
C 18:2 n6	4.61	4.72	4.56	0.39	0.081
C 18:3 n3	0.91	0.98	0.87	0.12	0.098
C 20:4 n6	0.51 ^b^	0.72 ^a^	0.49 ^b^	0.11	0.032
PUFA n6/n3	5.84 ^b^	5.74 ^b^	6.02 ^a^	0.21	0.044
AI ^1^	0.52 ^ab^	0.49 ^b^	0.57 ^a^	0.02	0.041
TI ^2^	1.30 ^ab^	1.21 ^b^	1.41 ^a^	0.05	0.045
h/H ^3^	2.05 ^ab^	2.17 ^a^	1.87 ^b^	0.07	0.039

^a, b^—values in the same rows with different letters differ significantly at *p* ≤ 0.05; ^1^ AI—atherogenicity index; ^2^ TI—thrombogenicity index; ^3^ h/H—Hypocholesterolemic/Hypercholesterolemic ratio; SFA—sum of saturated fatty acids; MUFA—sum of monounsaturated fatty acids; PUFA—sum of polyunsaturated fatty acids.

**Table 9 animals-13-01331-t009:** Fatty acid composition (% of all estimated fatty acid) and indices of fat dietetic value in backfat adipose tissue.

Fatty Acid	Groups/Cereal			SEM	*p* Value
	I—Barley	II—Triticale	III—Rye		
SFA	44.99 ^ab^	43.47 ^b^	45.74 ^a^	1.32	0.031
C 16:0	26.37 ^a^	24.97 ^b^	26.93 ^a^	0.82	0.042
C 18:0	16.63	16.54	16.72	0.44	0.108
MUFA	41.39 ^ab^	42.54 ^a^	40.72 ^b^	1.21	0.038
C 18:1 n9	36.03 ^ab^	37.12 ^a^	35.42 ^b^	0.47	0.036
C 18:1 n7	2.14	2.16	2.12	0.05	0.364
PUFA	13.27	13.61	13.13	0.23	0.117
C 18:2 n6	12.23	12.51	12.15	0.31	0.099
C 18:3 n3	0.78	0.81	0.75	0.03	0.139
C 20:4 n6	0.19	0.21	0.17	0.01	0.178
PUFA n6/n3	16.01	15.80	16.51	0.37	0.074
AI ^1^	0.59	0.55	0.62	0.04	0.057
TI ^2^	1.52	1.42	1.57	0.17	0.053
h/H ^3^	1.84	2.00	1.78	0.08	0.061

^a, b^—values in the same rows with different letters differ significantly at *p* ≤ 0.05; ^1^ AI—atherogenicity index; ^2^ TI—thrombogenicity index; ^3^ h/H—Hypocholesterolemic/Hypercholesterolemic ratio; SFA—sum of saturated fatty acids; MUFA—sum of monounsaturated fatty acids; PUFA—sum of polyunsaturated fatty acids.

## Data Availability

The data supporting reported results are in the possession of the Authors (E.R.G.; E.K.-V.; G.S.).

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
