# Peer review of "Barley, Triticale, or Rye? The Type of Grain Can Affect the Growth Performance and Meat Quality of Sustainable Raised Pigs"

_animals, 2023, doi:10.3390/ani13081331_

Round 1

Reviewer 1 Report

The manuscript “Barley, triticale, or rye? The type of grain can affect the growth performance and meat quality of organically raised pigs” is interesting and present-day (organic farming, soybean dietary partially/ totally replacement with other sources/ingredients).

 There are some minor observations that need to be clarified.

Introduction: it is comprehensive and clear, and draws attention to the specific topic of research

Materials and Methods:

Row 135: Please mention the producer or the origin of the organic cereals (national or international)

Row: 162: an average body weight within each pen- Please mention the average weight within brackets

Row: 211: Please add the authors of [27]

Row 222-223: “between 0.05 and 0.10 was considered a trend/tendency”. From my point of view ‘Trend’ signify some results towards significance, but is misleading.

In discussions or conclusions, please add some more comparisons or similar studies realized in conventional farming conditions on pigs, with the same type of cereals, more or less.

Author Response

The authors are very grateful for all valuable suggestions. We have tried to consider all comments and their explanations have been used to improve our manuscript.

Materials and Methods: Row 135: Please mention the producer or the origin of the organic cereals (national or international).

Answer: The producers of cereal grains were domestic (Poland) individual farmers, running their own ecological farms. This is added in the text (L 109).

Row: 162: an average body weight within each pen- Please mention the average weight within brackets.

Answer: it is done (L 166)

Row: 211: Please add the authors of [27]

Answer: it is done - Kozubek (1985)

Row 222-223: “between 0.05 and 0.10 was considered a trend/tendency”. From my point of view ‘Trend’ signify some results towards significance, but is misleading.

Answer: A trend/tendency is commonly used in statistical calculations to emphasize that the difference between the means is close to significance. It is accepted by editorials of scientific journals. In the description of the differences between the groups, the P value is given, which distinguishes significant results from trends/tendency. A good example would be a manuscript published in ANIMALS (https://doi.org/10.3390/ani13030381). To quote the authors: “A p < 0.05 was considered to be statistically significant, whereas 0.10 > p > 0.05 was considered a trend”. Indeed, we have settled the cutoff point for statistical significance at the level a=95%. However, we have not only analysed the results regarding the p-value (as we know the size of a p-value depends critically on the sample size) but also taken into consideration the CI values, so based on the results we believe we could prove the significance on a larger size of the sample.

In discussions or conclusions, please add some more comparisons or similar studies realized in conventional farming conditions on pigs, with the same type of cereals, more or less.

Answer: Actually, we had already mentioned in the Discussion (lines 338-343), that the literature data concerning influence of cereals from organic production on growth performance, carcass traits, meat and fat quality of fattening pigs is sparse due to the low availability of cereals from this type of cultivation. Thus, a great part of the Discussion is based on the results of studies where varied type of cereals was used, however obtained from conventional cultivation. In the Conclusion we would rather not like to mention it, because this chapter concerns the results of our experiment only.

Reviewer 2 Report

The paper is congruent with the special issue "Effects of Feed Ingredients on Growth Performance and Carcass Characteristics in Animals". However, the authors will have to completely re-written the paper: it is not, in fact, a biological trial. As shown in table 1, AAs of synthetic origin were used, which are not allowed by the organic method.

When the paper has been rewritten, it will certainly be resubmitted for the special issue.

I add  specific comments:

pag 4 - table 1 calcium carbonate instade fodder chalk

page 4 - line 158 specify how the content of meat in the carcass was calculated

page 5 - ileal digestibility paragraph: specify how long the animals were fasting and if fasting may have influenced the result

page 5 - line 187: specify which physicochemical properties were determined

pag 6 - line 223. pen should be used as an experimental unit also for body weight and average daily gain

page 8 table 5 - report only average backfat thickness; delete the 5 measurements 

page 9 table 7 - explain the abbreviations G-H and M/T

page 10 - Discussion . this paragraph needs to be greatly shortened and re-written (no organic trial) . Some parts are more suitable for introduction rather than discussion. 

Author Response

The authors are very grateful for all valuable suggestions. We have tried to consider all comments and their explanations have been used to improve our manuscript.

The authors will have to completely re-written the paper: it is not, in fact, a biological trial. As shown in table 1, AAs of synthetic origin were used, which are not allowed by the organic method. When the paper has been rewritten, it will certainly be resubmitted for the special issue.

Answer: As suggested, the manuscript was rewritten. The term "organic" was actually not properly used in our description. Thank you for pointing out this shortcoming of our manuscript.

pag 4 - table 1 calcium carbonate instade fodder chalk

Answer: it is done.

page 4 - line 158 specify how the content of meat in the carcass was calculated

Answer: it is done (L 160-162)

page 5 - ileal digestibility paragraph: specify how long the animals were fasting and if fasting may have influenced the result

Answer: This specification is added (L 174-175). Animals were not fasted prior to slaughter and collection of the sampling of intestinal digesta, as fasting significantly reduces its amount. The collection of digesta from unfasted animals allowed to avoid of it’s possible effect on the obtained results. In our experiment all animals were treated the same, so the effect of this factor on all animals was similar.

page 5 - line 187: specify which physicochemical properties were determined

Answer: it is done (L 192-193).

pag 6 - line 223. pen should be used as an experimental unit also for body weight and average daily gain

Answer: In the case of this experimental set-up, it seems reasonable to us to define the investigated traits as precisely as possible. In the case of pigs, both the body weight and the growth rate of an individual animal were determined (this specification is added in the M&M - chapter 2.2 and 2.7), so relying on pens in the statistical calculations of this parameters may offset the effect.

page 8 table 5 - report only average backfat thickness; delete the 5 measurements 

Answer: it is done.

page 9 table 7 - explain the abbreviations G-H and M/T

Answer: it is done.

page 10 - Discussion. this paragraph needs to be greatly shortened and re-written (no organic trial). Some parts are more suitable for introduction rather than discussion.

Answer: The Discussion paragraph has been shortened, the one part has been moved to the Introduction chapter, redundant words and overly detailed descriptions have also been removed.

Reviewer 3 Report

Dear authors, thank you for inyeresting work you have done. Although your experiment is done in the context of organic production, most of the available literature on the subject has been done with conventionally raised pigs.  For this reason, and to make your discussion more contextualized as well, perhaps a thesis of pigs fed corn, the most widely used and studied cereal for feeding pigs, would have been helpful.

There are two unclearpoints  in 2.3 Tissue collection in relation to the reported results. By what method was the lean meat of the carcass and ham calculated? I can imagine that a carcass scanning tool was used that also returns the amount of lean and fat of each commercial cut as well: in any case it would be better to indicate the classification devices used and the relative authorized method.

Author Response

The authors are very grateful for all valuable suggestions. We have tried to consider all comments and their explanations have been used to improve our manuscript.

Although your experiment is done in the context of organic production, most of the available literature on the subject has been done with conventionally raised pigs.  For this reason, and to make your discussion more contextualized as well, perhaps a thesis of pigs fed corn, the most widely used and studied cereal for feeding pigs, would have been helpful.

Answer: Due to the length of the manuscript and the other Reviewers suggestions we had to rewritten this paragraph and shortened it. We are unable to relate the results more extensively to maize-based pig feeding yet, but some of the results have been compared with those obtained when maize-based mixtures were fed to pigs, e.g. publication by McGhee and Stein (2020), Sullivan et al. (2007), Kim et al. (2014). Diets with high proportion of corn were not used in our study as such kind of feeding is not popular in Poland for extensively feeding pigs. High proportion of corn in the fatteners diet should be limited due to its significant negative impact on the physico-chemical quality of meat and lard.

There are two unclearpoints  in 2.3 Tissue collection in relation to the reported results. By what method was the lean meat of the carcass and ham calculated? I can imagine that a carcass scanning tool was used that also returns the amount of lean and fat of each commercial cut as well: in any case it would be better to indicate the classification devices used and the relative authorized method.

Answer: The content of lean meat in the carcass was determined using the Ultra FOM 300 apparatus (SFK Technology A/S, Herlev, DK) within 30 minutes from the start of slaughter operations – this information is also added in the text (L 160-162).

Round 2

Reviewer 2 Report

the paper can be published in its current form